# Exploring the Susceptibility of C3H Mice to Tick-Borne Encephalitis Virus Infection: Implications for Co-Infection Models and Understanding of the Disease

**DOI:** 10.3390/v15112270

**Published:** 2023-11-17

**Authors:** Stefania Porcelli, Aurélie Heckmann, Anne-Claire Lagrée, Clémence Galon, Sara Moutailler, Pierre Lucien Deshuillers

**Affiliations:** Laboratoire de Santé Animale, ANSES, INRAE, Ecole Nationale Vétérinaire d’Alfort, UMR BIPAR, F-94700 Maisons-Alfort, France; stefania.porcelli@anses.fr (S.P.); aurelie.heckmann@anses.fr (A.H.); anne-claire.lagree@vet-alfort.fr (A.-C.L.); clemence.galon@anses.fr (C.G.)

**Keywords:** tick-borne encephalitis virus, C3H mice, mouse model, tick

## Abstract

Ticks and tick-borne diseases (TBDs) are increasingly recognized as a critical One Health concern. Tick-borne encephalitis (TBE), a severe neuro infection caused by the tick-borne encephalitis virus (TBEV), has emerged as a significant global public health threat. Laboratory animals, particularly mice, have played a pivotal role in advancing our understanding of TBD pathogenesis. Notably, BALB/c mice have been employed as models due to their heightened susceptibility to TBEV. However, the use of C3H mice, valued for other tick-borne pathogens, has remained unexplored for TBEV until now. This study aimed to assess the susceptibility of C3H mice to TBEV infection, laying the groundwork for future co-infection models involving TBEV and *Borrelia*. Experiments revealed that C3H mice are susceptible to TBEV infection through subcutaneous inoculation. While 10^2^ PFU/mouse appeared necessary for full infection, 10^3^ PFU/mouse induced consistent symptoms. However, subsequent assessment of ticks’ acquisition of TBEV from infected mice met with limited success, raising questions about optimal infectious doses for natural infection. These findings suggest the potential of C3H mice for studying TBEV and co-infections with other pathogens, particularly *Borrelia*. Further exploration of the interplay between these pathogens, their transmission dynamics, and disease severity could enhance prevention and control strategies.

## 1. Introduction

Ticks and tick-borne diseases (TBDs) are currently a growing One Health issue [1]. Tick-borne encephalitis (TBE), one of the most dangerous neuro infections, has become an increasingly significant public health concern in Europe and other parts of the world [2,3]. It is caused by the tick-borne encephalitis virus (TBEV), a member of the Flavivirus genus (Flaviviridae family). The fatality rate varies between TBEV subtypes: 1% to 2% for the European subtype (TBEV-Eu), 6% to 8% for the Siberian subtype (TBEV-Sib), and 20% to 30% for the far-eastern subtype (TBEV-FE) [4].

TBEV mainly spreads between ticks and certain animals like deer and small rodents. Although it occasionally infects large animals like sheep, cattle, and humans, they are accidental hosts, and the virus remains in their blood only briefly, with low or undetectable levels. Consequently, these large animals do not effectively transmit the virus [5]. In Europe, small rodents such as the bank vole (*Myodes glareolus*) and yellow-necked mouse (*Apodemus flavicollis*) play a crucial role as a bridge host for the virus between the different life stages of the tick through a process known as co-feeding. Ticks can also become infected by feeding on rodents with the virus in their bloodstream (viremic hosts). However, the primary route of infection in the ecosystem is believed to be tick co-feeding on non-viremic or immune rodents, particularly for the European subtype of the virus (TBEV-Eur) [6].

The transmission of TBEV-Eu to humans occurs through the bite of an infected tick, *Ixodes ricinus* [4]. Following a tick bite, the skin becomes the primary site of viral replication. Various cell types such as Langerhans cells, keratinocytes, dermal macrophages, and neutrophils recruited during the inflammatory response to the tick bite [7] are responsible for carrying the viral particles to nearby lymph nodes. The virus replicates in the lymph nodes, leading to its dissemination into the bloodstream, causing viremia. The virus can then infect peripheral tissues like the spleen, liver, and bone marrow through the lymphatic and circulatory systems. During this stage, the virus crosses the blood–brain barrier and infects the brain. However, the exact mechanism of this process remains unclear. Fortunately, most infected patients can eliminate the virus through immune mechanisms, primarily involving cytotoxic lymphocytes [4,8].

The biphasic nature of TBE arises from the virus initially spreading in peripheral tissues, triggering a cytokine response. In some instances, the virus penetrates the central nervous system (CNS), leading to the establishment of a second neurological phase of the disease. After an incubation period of 7–10 days, about 70% of patients experience non-specific flu-like symptoms during the viremic phase [9]. However, in 20–30% of cases, the neurological phase can manifest as meningitis, meningoencephalitis, or meningoencephalomyelitis in approximately 50%, 40%, and 10% of patients, respectively [10].

Laboratory animals have played a crucial role in enhancing our comprehension of the pathogenesis of TBDs. Mice are commonly used as animal models for flavivirus encephalitis, primarily because laboratory strains are highly susceptible to the disease and exhibit symptoms and viral infection patterns similar to those of humans [11]. IFNAR^−/−^ and BALB/c mice are often employed as models for TBEV infection due to their high susceptibility to the virus. IFNAR^−/−^ mice are particularly vulnerable and typically do not survive beyond 6 days post infection, while BALB/c mice generally survive for 11 days before succumbing to the virus [12,13]. Conversely, the use of C3H/HeN (C3H) mice to study TBEV infection has not been described to our knowledge, although they have been used to study other tick-borne pathogens (TBPs). Indeed, C3H mice are highly susceptible to *Borrelia* infection and tend to develop Lyme arthritis, unlike BALB/c mice [14,15]. Therefore, C3H mice are commonly used to study *B. burgdorferi* sensu stricto (s.s.) and other TBPs such as *Anaplasma phagocytophilum* or *Babesia microti* in addition to co-infection between TBPs or transmission of TBPs by *Ixodes scapularis* ticks [15,16,17].

Thus, our study was designed to assess the susceptibility of C3H mice to TBEV infection in order to establish a new co-infection model in the future with TBEV and *B. afzelii*, and explore interactions between different pathogens in co-infected mice and ticks.

Two separate experiments were conducted using C3H mice. In the first experiment, the aim was to (i) identify the optimal dose (PFU/mouse) of TBEV to administer to mice subcutaneously in order to detect the virus in blood samples at different time points and (ii) confirm that this dose could effectively infect the brains of infected mice. Based on the results of this first experiment, a second one was conducted to (i) establish the optimal infectious dose necessary to induce TBEV symptoms in mice, assess virus distribution in the blood, and measure the amount in the organs, and finally, (ii) determine the ability of ticks to acquire TBEV from infected mice and maintain the virus after molting.

## 2. Materials and Methods

Two distinct experiments entailing the infection of C3H mice with TBEV were performed. In each experiment, our objective was twofold: (i) to determine the optimal dose for detecting TBEV from blood samples collected on different days, and (ii) to ensure that this optimal dose could successfully infect the brains of susceptible mice in the related group (definitive proof of infection with TBEV of susceptible mice). In the second experiment, our objectives were to (i) establish the optimal infectious dose needed to induce TBEV symptoms in mice, assess virus distribution in the blood, and measure the amount in the organs, and finally, (ii) determine the ability of ticks to acquire TBEV from infected mice and maintain the virus after molting.

### 2.1. Experimental design

The experimental design of this study is presented in Figure 1.

In the first experiment, after one week of acclimatization, three groups of mice (*n* = 5 in each group) were inoculated subcutaneously with 100 μL of a viral suspension (first viral production) corresponding to 1, 10, and 10^2^ PFU/mouse, respectively, while five control mice were inoculated with 100 μL of Dulbecco’s modified Eagle’s medium (DMEM). Symptoms were recorded every day for 11 days.

Blood samples (30 µL) were collected from each mouse on days 0, 4, 6, 8, and 11. Brains were collected following death related to TBEV or 11 days post infection when mice still alive were sacrificed. All samples were stored at −80 °C until further use.

In the second experiment, twelve six-week-old mice (seven mice for 10^2^ PFU and five mice for 10^3^ PFU) were infected with TBEV (second viral production) through subcutaneous inoculation with 100 μL of the virus at two different dilutions (10^2^ PFU/mouse, 10^3^ PFU/mouse), while two additional mice were used as a negative control, being inoculated with 100 μL of DMEM. Blood samples (30 µL) were collected on days 0, 4, 11, and 15 in all mice.

Symptoms and death were recorded for each mouse for 15 days. Some of the mice had to be euthanized before the endpoint due to severe clinical signs of distress due to TBE, and their organs were collected. The remaining live mice were sacrificed on day 15 post infection and different organs (brain, spleen, heart, kidney, liver, lung) were then collected. The samples were stored at −80 °C until further use.

For eleven mice (six mice for 10^2^ PFU, four for 10^3^ PFU, and one mouse as a negative control), naive larvae (*n* = 100) were placed in a capsule system on the back of each mouse at different time points (days 4, 5, 6 or 8), to feed until repletion (maximum 3 days of feeding) as described in Migné et al. [18]. Once fully engorged, the larvae were placed in a desiccator for storage. On the same day, 20 engorged larvae from the TBEV groups and 10 engorged larvae from the negative group were sacrificed and conserved at −80 °C until further use. The remaining ticks were put back in the desiccator until molting. After the molt (around one month later), 30 nymphs from the 10^2^ PFU group and 20 nymphs from the 10^3^ PFU group were sacrificed and conserved at −80 °C until further use.

### 2.2. Mice

Six-week-old female C3H/HeN mice were obtained from Charles River Laboratories (Calco, Italy). For the first experiment, the mice were in a group of five. In the second experiment, a group of two mice was used as the negative control, while seven mice were infected with 10^2^ PFU per mouse, and a group of five mice were infected with 10^3^ PFU. When infested with ticks, the mice were kept individually in plastic cages with wood-chip bedding, fed ad libitum, and kept in standardized conditions (21 °C, 12:12 light/dark cycle) at the animal facilities. The present research was conducted in compliance with all relevant European Union guidelines for work with animals and with the French national legislation on the use of animals for experimentation and protection against cruelty. Experimental protocols were approved by the ANSES-ENVA-UPEC Ethics Committee for Animal Experimentation (Approval Number: APAFIS#31496-2021051213296446 v3, 18 June 2021).

### 2.3. Ticks

*I. ricinus* larval ticks were derived from the 1st generation of ticks collected in the Senart Forest (Épinay-sous-Sénart, France). Female ticks were tested for TBEV RNA after laying eggs to ensure that they did not carry the virus before being used in the experiment. The ticks were maintained in the laboratory at a temperature of 21 °C with 95% humidity in a desiccator and exposed to a 12 h light/12 h dark photoperiod. Experimental protocols for the tick colony were approved by the ANSES-ENVA-UPEC Ethics Committee for Animal Experimentation (Approval Number: APAFIS #35511-2022022111197802 v2, 14 April 2021).

### 2.4. Virus

Mice were infected with TBEV strain Hypr. This strain, known to be highly neuroinvasive and neurovirulent for laboratory mice, was isolated from human blood in the Czech Republic in 1953 [19]. TBEV Hypr was passaged four times in suckling mice brains and twice in Vero E6 cells before being used in the experiments. Two different viral productions were used in the experiment. The first one had a titer of 2.3 × 10^6^ PFU/mL, while the second had a titer of 5 × 10^6^ PFU/mL.

### 2.5. RNA Extraction

RNA was extracted from each blood sample (30 µL) using the NucleoSpin^®^ RNA extract II kit (Macherey Nagel, Düren, Germany) following the manufacturer’s instructions. The RNA samples were eluted in 40 µL of RNase-free water and stored at −80 °C until further use.

Before extracting RNA from each organ (½ brain, ½ spleen, ½ heart, 1 kidney, ½ liver, ¼ lung), they were individually homogenized in 500 μL of DMEM supplemented with 10% fetal bovine serum (FBS) using the homogenizer Fast Prep-24 5G (MP Biomedicals, Irvine, CA, USA) with six stainless steel beads. The homogenization was performed with three cycles at 5500× rpm for 20 s and centrifuged for 5 min at 1500× *g* to ensure that the organs were thoroughly disrupted and mixed with the medium. RNA was then extracted from 250 µL of the supernatant using the NucleoSpin^®^ RNA extract II kit following the manufacturer’s instructions. The RNA samples were eluted in 40 µL of RNase-free water and stored at −80 °C until further use.

Ticks were homogenized individually in 350 µL of lysis buffer along with 3.5 µL of β-mercaptoethanol. The homogenization process was carried out using the same homogenizer Fast Prep-24 5G (MP Biomedicals, USA) with six stainless steel beads for two cycles at 5500× rpm for 20 s. After homogenization, the samples were centrifuged for 5 min at 1500× *g*. RNA was then extracted from the supernatant, eluted in 30 µL of RNase-free water, and stored at − 80 °C until further use.

### 2.6. Real-Time RT-PCR

The real-time RT-PCR was performed using the LightCycler^®^ 480 RNA Master Hydrolysis Probes kit (Roche Diagnostics, Mannheim, Germany) following the manufacturer’s instructions.

For the experiment, a 20 µL reaction volume was used for the RT-PCR mix. This was obtained by combining 7.4 µL of LightCycler^®^ 480 RNA Master Hydrolysis Probes, 8.05 µL of water, 1.3 µL of activator, 0.5 µL of primers, and 0.25 µL of probes to target TBEV, and 2 µL of RNA template.

The following primers and probe were used: TBE_euro_F: TCC TTG AGC TTG ACA AGA CAG (final concentration: 0.5 µM), TBE_Euro_R: TGT TTC CAT GGC AGA GCC AG (final concentration: 0.5 µM), and TBE_Euro_P: TGG AAC ACC TTC CAA CGG CTT GGC A (final concentration: 0.25 µM) [20]. The real-time RT-PCR parameters consisted of a reverse transcription cycle at 63 °C for 3 min, followed by a denaturation cycle at 95 °C for 30 s; 45 successive cycles at 95 °C for 10 s, 60 °C for 30 s, and 72 °C for 1 s; and a final step of cooling at 40 °C for 30 s.

### 2.7. Reverse Transcription (RT) and cDNA Preamplification

The cDNA from mice organs was prepared using the Reverse Transcription Master Mix (Standard Biotools, San Francisco, CA, USA) and then pre-amplified using the Preamp Master Mix (Standard Biotools, San Francisco, CA, USA) as per the manufacturer’s instructions.

For the Reverse Transcription Master Mix, a 5× master mix was used, which included 1 µL of Reverse Transcription Master Mix, 3 µL of RNase-free water, and 1 µL of RNA. The reverse transcription process involved a thermocycler with cycles at 25 °C (5 min), 42 °C (30 min), and 85 °C (5 min). After the cDNA samples were obtained, they were either stored at −20 °C or used immediately for preamplification reactions using the Preamp Master Mix Kit. In the preamplification step, primers targeting TBEV had a final concentration of 0.2 μM each. The preamplification reactions were conducted in a final volume of 5 μL, comprising 1 µL of Preamp Master Mix, 1.25 μL of the pooled primer mixture, 1.5 µL of distilled water, and 1.25 μL of cDNA.

The PCR run conditions involved an initial cycle at 95 °C (2 min), followed by 14 amplification cycles at 95 °C (15 s) and 60 °C (4 min). Finally, the pre-amplified products were diluted with various dilutions (1:2, 1:5, 1:100, 1:500, 1:1000) using distilled water and kept at −20 °C for future use.

### 2.8. Digital PCR on the Biomark^TM^ System 

Digital PCR (dPCR) is a technique that divides a single sample into multiple individual PCR reactions. In this study, dPCR amplifications were conducted on mice organs using the Biomark^TM^ System with qdPCR 37k IFC digital array microfluidic chips (Standard Biotools, San Francisco, CA, USA). The Biomark^TM^ Integrated Fluidic Circuit (IFC) controller utilized nanoscale valves and channels to partition each of the 48 samples, pre-mixed with PCR reagents, into a panel of 770 PCR reaction chambers, resulting in a total of 36,960 individual qdPCR reactions on the digital array. The number of target molecules in each sample was accurately estimated by counting the positive reactions, based on the Poisson distribution [21].

Each reaction mixture consisted of a total volume of 6 µL comprising 3 µL of 2× Perfecta qPCR tough mix along with ROX reference dye from Standard Biotools, 0.6 µL of 20× GE Sample Loading Reagent also from Standard Biotools, 0.3 µL of 20× primer stock containing 100 μM concentration each of forward and reverse primers, 20 μM of TBEV probe, and 1.8 µL of pre-amplified cDNA sample. The experiments included positive controls containing cDNA extracted from a virus suspension and negative controls containing water.

Out of the 6 µL of reaction mix, 4 µL was used in loading the chip with the IFC controller MX, and 0.65 µL was actually partitioned into the 770 chambers of one panel, including 0.38 µL of cDNA extract. The dPCR program consisted of 2 min at 50 °C, 10 min at 95 °C, and then 40 cycles during which the samples were subjected to 95 °C for 15 s and 60 °C for 60 s. Fluorescence was recorded by the apparatus at the end of each elongation step (1 min at 65 °C) for every amplification cycle. The Digital PCR Analysis software, a component of the Biomark system (Standard Biotools), was used to count the number of positive chambers out of the total number of chambers per panel. The Poisson distribution was used to estimate the average number of template copies per chamber in a panel. Each sample was characterized by its corresponding absolute quantity, and no positive chambers were observed in negative samples.

## 3. Results

### 3.1. Susceptibility of C3H Mice to TBEV

We attempted to evaluate the susceptibility of C3H mice to TBEV (strain Hypr).

In the first experiment, three different doses of TBEV were inoculated subcutaneously into batches of five mice.

To examine the presence of TBEV in the mice, we detected viral RNA from blood samples (D0, D4, D6, D8, D11) and from the brain. In the TBEV groups infected with 1 PFU and 10 PFU, viral RNA was observed in the blood of only four out of five mice on days 4, 6, and 8 post infection (p.i.).

Additionally, in both groups, one mouse remained negative for TBEV at all the different blood collection time points. Viral RNA was detected in the blood of all five mice infected with TBEV 10^2^ PFU on days 6 and 8 p.i. By day 11 p.i., viral RNA was found in the blood of all four remaining mice, as one mouse had already died on day 8. Conversely, on day 4 p.i., viral RNA was only detected in the blood of four mice (Table 1).

Among the mice infected with 1 PFU of TBEV, two mice succumbed on day 11 p.i. In the TBEV group infected with 10 PFU, one mouse died on day 10 p.i. and another on day 11 p.i. However, in the TBEV group infected with 10^2^ PFU, only one mouse died, on day 8 p.i.

The presence of TBEV RNA in the brains of the mice was detected through real-time RT-PCR. TBEV RNA was detected in the brains of all five mice that received 10^2^ PFU, whereas TBEV was detected only in four out of five brains from mice inoculated with 1 PFU and 10 PFU (Table 2).

Once it was established that the optimal dose for achieving a 100% infection rate in mice was 10^2^ PFU (100% positive brains and 100% positive blood), we conducted a second experiment. This experiment was designed to evaluate: (i) the clinical symptoms, (ii) the dissemination of the virus in the blood and its quantification in different organs, and (iii) the ability of *I. ricinus* ticks to acquire the virus during blood meals. This assessment involved testing two doses: 10^2^ PFU and 10^3^ PFU using C3H mice.

A characteristic dose–response curve was observed for TBEV-infected mice with 10^2^ and 10^3^ PFU. Infected mice began to exhibit TBEV symptoms between 8 and 14 days p.i. Symptoms in both groups included ruffled fur, a hunched posture, and paralysis.

We observed that the abdomens of three out of seven mice infected with 10^2^ PFU and four out of five infected with 10^3^ PFU became swollen before the mice became moribund and were euthanized.

All of the mice (5/5) in the TBEV 10^3^ PFU group died between 10 and 13 days post infection. Similarly, in the TBEV 10^2^ PFU group, six out of seven mice died within a time frame of 10 to 12 days p.i. (Figure 2). Nevertheless, in this group, one mouse remained asymptomatic and survived without succumbing to the infection.

To investigate viral RNA in the mice, we collected blood samples at various time points and detected the virus using real-time RT-PCR.

On days 4 and 11 p.i., all blood samples from mice inoculated with 10^3^ PFU tested positive for TBEV RNA. In contrast, among the mice in the 10^2^ PFU group, one out of seven showed negative results for virus detection in their blood at different time points (Table 3).

Among the mice inoculated with 10^3^ PFU, all five tested positive for viral RNA in their brains (100%). In contrast, in the group inoculated with 10^2^ PFU, the percentage of brains infected was 85% (6/7). In this group, only one mouse tested negative for viral RNA in the brain, the same one that tested negative in the blood, too.

The brain had the highest TBEV RNA levels in both groups, with around 1.8 × 10^6^ copies/µL (10^2^ PFU group) and 1.3 × 10^6^ copies/µL (10^3^ PFU group). The heart showed medium levels—about 1.3 × 10^5^ copies/µL (10^3^ PFU group) and 2.5 × 10^5^ copies/µL (10^2^ PFU group)—while the lungs had similar levels at approximately 1.4 × 10^5^ copies/µL (10^3^ PFU group) and 2 × 10^5^ copies/µL (10^2^ PFU group). In contrast, the liver, kidney, and spleen had lower TBEV RNA levels ranging, from 1.4 × 10^3^ to 3.6 × 10^3^ copies/µL in the 10^3^ PFU group and from 5.9 × 10^2^ to 6.3 × 10^2^ copies/µL in the 10^2^ PFU group (Figure 3).

However, no significant differences were observed between the two different doses used regardless of the organs tested.

### 3.2. TBEV Acquisition by Naive Larvae from Infected Mice

In the latter phase of the second experiment, we placed naive larvae (*n* = 100 per mouse; 2 mice per group) at different time points (days 5, 6, or 8 in the 10^2^ PFU group; days 4 or 5 in the 10^3^ PFU group; day 4 in the negative control group) to evaluate their ability to: (1) acquire the virus and (2) conserve the virus after molting (i.e., transstadial transmission).

Viral RNA was detected in engorged larvae and nymphs (after molting) through real-time RT-PCR. In the group of mice inoculated with 10^2^ PFU, viral RNA was detected in 25% (5/20) of engorged larvae placed on mice on day 5 p.i., in 10% (2/20) for the ones placed on day 6, and not detected (0% (0/20)) for the one placed on day 8 p.i. Conversely, in the group of mice inoculated with 10^3^ PFU, viral RNA was detected in 10% of engorged larvae (2/20) placed on mice on day 4 and in 35% (7/20) in the ones placed on day 5 p.i. For all groups, engorged larvae that tested positive exhibited a late CT value. None of the nymphs tested after molting were positive, so no transstadial transmission from infected engorged larvae to nymphs after molting could be evidenced (Table 4).

## 4. Discussion

Our study was designed to assess the susceptibility of C3H mice to TBEV infection, to define the minimal infectious dose needed for reproducible results, to assess symptoms and targeted organs following TBEV infection, and finally, to assess the acquisition of TBEV by ticks during blood meals. We have demonstrated that C3H mice are susceptible to TBEV infection through subcutaneous inoculation, with paralysis and death occurring between 10 and 13 days post infection. Moreover, even if a minimum dose of 10^2^ PFU/mouse seems necessary to induce 100% infected mice (100% infected blood and brain), 10^3^ PFU/mice seems to be the optimal dose to obtain reproducible symptoms in mice and to infect engorged *I. ricinus* larvae. Finally, both the doses of 10^2^ and 10^3^ PFU/mice induce infection of all the tested organs (brain, heart, lung, liver, kidney, spleen), with the brain, heart, and lungs containing the highest amount of viral RNA in both cases. Nevertheless, it is important to note that we did not backtitrate the virus inocula. So, we could not be sure that the results obtained in groups with a very low virus dose were not due to an incorrect estimation of the number of PFUs per inocula. To avoid this situation, backtitration of the inocula should be performed in future experiments.

IFNAR-deficient (IFNAR^−/−^) and BALB/c mice are often employed as models for studying TBEV infection due to their high susceptibility to the virus. BALB/c mice are subjected to subcutaneous infection with a dose of 10^3^ PFU, while IFNAR^−/−^ mice undergo intraperitoneal infection with a dose of 10^4^ PFU. Both of these doses result in a 100% mortality rate.

IFNAR^−/−^ mice are particularly vulnerable and typically do not survive beyond 6 days post infection, while BALB/c mice generally survive for 11 days before succumbing to the virus [14,15].

However, the use of C3H/HeN (C3H) mice to study TBEV infection had not previously been described to our knowledge, even though they have been used for research on other TBPs. Indeed, C3H mice are highly susceptible to *Borrelia* infection and tend to develop Lyme arthritis, unlike BALB/c mice [14,15]. Therefore, C3H mice are commonly used to study *B. burgdorferi* s.s., other TBPs, co-infection between *Borrelia* and *Anaplasma* and the acquisition of these pathogens by *I. scapularis* ticks [15,16,17]. Thus, the fact that this study has been able to demonstrate the susceptibility of C3H mice to TBEV means that they constitute a new model of major interest for the study of TBP interactions.

Interestingly, during our first experiment, we noted that a few mice that received a low infectious dose (1 or 10 PFU) died earlier than those receiving a higher dose (10^2^ PFU). In fact, it has been acknowledged since the 1940s that mortality is not dependent on the dose of encephalitic flaviviruses [22,23], yet the reason remains unknown. Several factors can affect the severity of TBE, including but not limited to the amount of virus introduced into the body and its virulence [24], the host’s age and immune condition [25], as well as the host’s genetic background, which may affect susceptibility to the virus. Thus, CNS invasion by the virus is not the only factor that determines whether the mice will die or not.

Moreover, during our second experiment, after subcutaneously infecting C3H mice with 10^2^ and 10^3^ PFU, three out of seven mice infected with 10^2^ PFU and four out of five infected with 10^3^ PFU showed abdominal distension. These alterations of the gastrointestinal system were consistent with the study conducted by Boelke et al. on BALB/c mice intravenously infected with the Sofjin strain of TBEV, using doses ranging from 10 to 10^3^ PFU. The study demonstrated that the distention of the small intestine was caused by peripheral neuritis resulting from infection of the myenteric plexus, which is part of the enteric nervous system [2]. Contrary to CNS symptoms frequently observed in TBEV infections, reports of gastrointestinal (GI) symptoms are infrequent in humans. Versace et al. reported a case of a man infected with TBEV and suffering from encephalomyeloradiculitis who experienced significant intestinal motility problems resembling those associated with bowel obstruction [26]. Therefore, this animal model may mimic what happens in humans with TBEV infection.

The quantification of viral RNA using digital PCR following RT and preamplification steps in this study demonstrated the high amount of virus (10^6^ copies/µL) in the brains of mice infected with both 10^2^ and 10^3^ PFU, confirming the virus’s preference for the CNS. All other organs tested positive with high doses in the heart and lungs. Due to the tropism of the virus for the CNS, BALB/c and C57Bl/6 mice strains have become useful models to study the neuro pathogenicity and immune response of different strains of TBEV. According to our results, C3H mice could also play this role in such contexts [27,28,29]. Even if laboratory mice models are not a natural host of TBEV, they appear to mimic quite efficiently what happens in nature, as similar results have been found in *Microtus arvalis* (*M. arvalis*) infected subcutaneously with 10^3^ PFU of TBEV-Hypr. The same quantity of virus (10^6^ genome copies per mg of organ) was found in the brains of voles at 5 days p.i., and the researchers observed varying amounts of TBEV in the voles’ hearts, livers, spleens, and kidneys, suggesting that these organs were less affected [5]. However, it is important to consider that these wild animals have different genetic backgrounds than laboratory animals, which may influence the outcomes of the infection.

Finally, in our study, we observed that only a few engorged larvae were found positive via real-time PCR but there was no transstadial transmission from infected engorged larvae to nymphs after molting. The first hypothesis of this failure is that RNA copies of TBEV could be below the detection threshold in our infected ticks. Indeed, positive engorged larvae presented late Ct values, so maybe the small quantity of viral RNA in nymphs could not be detected. To test this hypothesis, we could homogenize nymphs and then inoculate this homogenate into newborn mice intracranially, potentially allowing us to detect even a few particles of viral RNA [21].

A second explanation could be that the tested infectious doses were insufficient to generate a significant viremia level in mice blood and thus did not allow the tick to acquire the virus during their blood meal. In nature, a specific study estimated the viral load in the blood of rodents in Russia: TBEV-FE RNA was measured at 2.4 × 10^5^ copies/mL, while TBEV-Sib RNA was observed at 2.4 × 10^2^ copies/mL [30], so we could hypothesize that such an amount is probably the approximate dose needed to infect ticks in their natural environment. Our study quantified 10^3^ copies/µL in mouse blood. Moreover, Belova et al. achieved successful infection of ticks by employing two strains of TBEV (Siberian and European subtypes). They infected mice with higher doses of the virus (10^5.8^ and 10^7^ PFU/mice) and obtained positive PCR results for TBEV detection by pooling three engorged larvae and nymphs. Another notable difference in their approach was the analysis of ticks at various time intervals after feeding on mice, considering the potential physiological state of the ticks during the assessment of infection rates [31].

Additionally, several artificial infection techniques have been developed and implemented to infect hard ticks with pathogens. For example, Migné et al. were able to infect 70% of engorged larvae with TBEV using 10^5^ PFU/mL in an artificial membrane feeding system [18]. The advantage of their system lies in the fact that we can directly control the concentration of pathogens in the blood that ticks consume [32]. Unfortunately, to the best of our knowledge, there is no study that has evaluated the minimal infection dose of TBEV in mouse blood (PFU, or viral RNA copies) that allows the acquisition and infection of ticks with TBEV. This type of study is needed.

In summary, we demonstrated that C3H mice are susceptible to TBEV infection with a minimal dose of 10^2^ PFU, and that a dose of 10^3^ is optimal to obtain infected *I. ricinus* ticks, but is probably not sufficient to allow transstadial transmission of the virus from engorged larvae to nymphs (after molting). Additional investigations using this strain of mice with a higher dose of the virus so that ticks become infected are needed to gain a better understanding of ticks’ acquisition of the virus in this particular model. While infecting mice with a higher dose is a possibility, it is crucial to consider the potential exacerbation of TBEV symptoms and mortality in mice, as such a protocol may deviate significantly from natural conditions.

## 5. Conclusions

This study has proven that C3H mice are susceptible to TBEV infection. As these mice are a good model for tick-borne pathogen infection, studies of co-infection with TBEV and other tick-borne bacteria such as *Borrelia* spp. are now possible. A better understanding of the interplay between TBEV and *Borrelia* could be gained by studying the effects of the immune response and the severity of both diseases in C3H mice. Future investigations should focus on assessing the success of these pathogens in their transmission from co-infected mice to non-infected ticks, as well as from co-infected ticks to non-infected mice. This research would contribute to our knowledge of the impacts of co-infection on disease dynamics and potentially aid in the development of more effective prevention and control strategies.

## Figures and Tables

**Figure 1 viruses-15-02270-f001:**
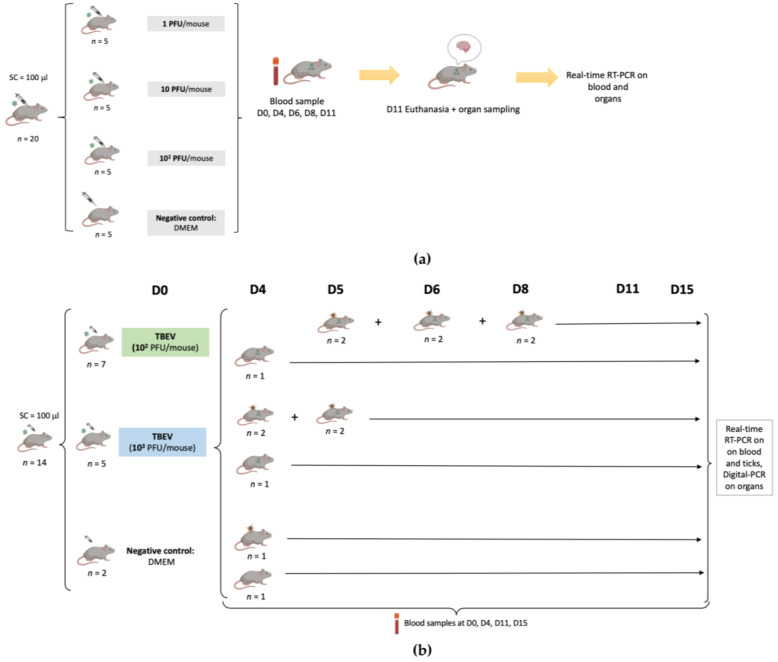
Representation of the experimental design describing the two experiments. (**a**): Infection of C3H mice with TBEV (first viral production) with 1 PFU/mouse, 10 PFU/mouse, 10^2^ PFU/mouse. Blood samples were collected on different days from the infected mice, and mice still alive were sacrificed on day 11 p.i. to collect their brain. Samples were analyzed using real-time RT-PCR. (**b**): Infection of C3H mice with TBEV (second viral production) with 10^2^ PFU/mouse, 10^3^ PFU/mouse. The infection was monitored by collecting blood samples on different days and mice still alive were sacrificed on day 15 p.i. to collect their organs. Acquisition of TBEV by ticks was evaluated by placing naive larvae on the backs of mice at different time points. Blood samples and ticks were analyzed using real-time RT-PCR, and organs were analyzed using digital PCR following RT and preamplification steps.

**Figure 2 viruses-15-02270-f002:**
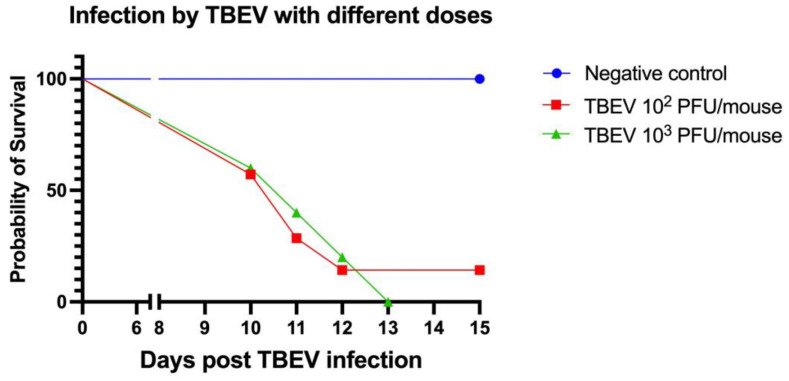
Experiment B. Survival curve of C3H mice inoculated subcutaneously with two doses of TBEV (strain Hypr): 10^2^ PFU (*n* = 7) and 10^3^ PFU (*n* = 5); and one negative group inoculated with medium (*n* = 2).

**Figure 3 viruses-15-02270-f003:**
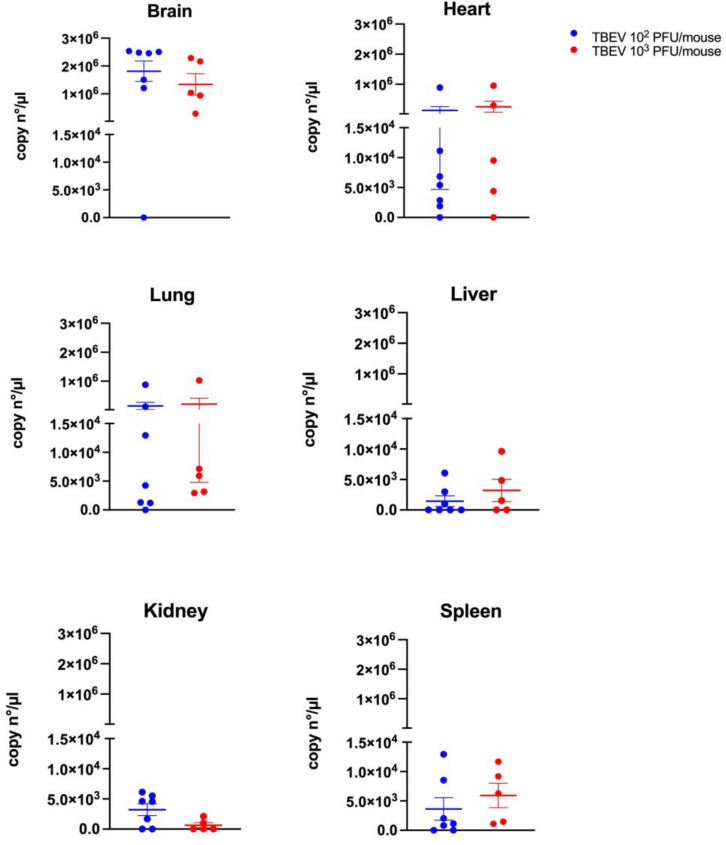
TBEV RNA quantification in the organs of C3H mice infected with 10^2^ and 10^3^ PFU using digital PCR following RT and preamplification steps on day 15 p.i. from mice that were euthanized, or between day 10 and day 13 p.i. for mice that died before. Means and standard errors are shown.

**Table 1 viruses-15-02270-t001:** Experiment A. TBEV RNA detection in the blood of C3H mice using real-time RT-PCR.

Days Post Infection	D0	D4	D6	D8	D11
Negative control	0/5 *	0/5	0/5	0/5	0/5
TBEV 1 PFU	0/5	4/5	4/5	4/5	2/3
TBEV 10 PFU	0/5	4/5	4/5	4/5	2/3
TBEV 10^2^ PFU	0/5	4/5	5/5	5/5	4/4

* The number of positive mice/numbers of tested mice. D = day.

**Table 2 viruses-15-02270-t002:** Experiment A. TBEV RNA detection in the brains of C3H mice using real-time RT-PCR.

Days Post Infection	Brain
Negative control	0/5 *
TBEV 1 PFU	4/5
TBEV 10 PFU	4/5
TBEV 10^2^ PFU	5/5

* The number of positive brains/number of mouse brains harvested on day 11 p.i. from euthanized mice. The organs of mice that died before day 11 were collected at the time of death.

**Table 3 viruses-15-02270-t003:** Experiment B. TBEV RNA detection in the blood of C3H mice using real-time RT-PCR.

Days Post Infection	D0	D4	D11	D15
Negative control	0/2 *	0/2	0/2	0/2
TBEV 10^2^ PFU	0/7	6/7	3/4	0/1
TBEV 10^3^ PFU	0/5	5/5	3/3	nd

* The number of positive mice/tested mice. D = day; nd = no data.

**Table 4 viruses-15-02270-t004:** Experiment B. TBEV RNA detection in (1) engorged larvae collected after feeding on C3H mice at different time points, and (2) in nymphs after molting.

	Engorged Larvae				Nymphs after Molt			
Days Post Infection	D4	D5	D6	D8	D4	D5	D6	D8
Negative control	0/10 *	nd	nd	nd	0/5	nd	nd	nd
TBEV 10^2^ PFU	nd	5/20 (25%)	2/20 (10%)	0/20	nd	0/10	0/10	0/10
TBEV 10^3^ PFU	2/20 (10%)	7/20 (35%)	nd	nd	0/10	0/10	nd	nd

* The number of positive ticks/total analyzed (percentage of infection %); D = day; nd = no data.

## Data Availability

Data are contained within the article.

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
