# Peer review of "Exploring the Susceptibility of C3H Mice to Tick-Borne Encephalitis Virus Infection: Implications for Co-Infection Models and Understanding of the Disease"

_viruses, 2023, doi:10.3390/v15112270_

Round 1

Reviewer 1 Report

Comments and Suggestions for Authors

The manuscript entitled “Exploring the Susceptibility of C3H Mice to Tick-Borne Encephalitis Virus Infection: Implications for Co-Infection Models and Understanding of the Disease” by Stefania et al. reported some very interesting findings.

Tick-borne encephalitis caused by tick-borne encephalitis virus is prevalent in many parts of Europe and Asia, and can cause severe neurological symptoms in humans. Understanding the susceptibility of different animal models to the infection is crucial for studying the disease and developing effective prevention and treatment strategies.

The findings of this report have important implications for the development of co-infection models and the understanding of TBEV pathogenesis.

The manuscript was well writte.

Author Response

We thanks the reviewer for his reviewing and his comment regarding our manuscript

Reviewer 2 Report

Comments and Suggestions for Authors

The manuscript by Stefania and colleagues is a very well written and presented description of evaluating C3H mice for susceptibility to a single strain of tick-borne encephalitis virus.  The study is quite small in scope, especially in terms of the number of mice used and lack of pathology, but overall, it provides a solid preliminary understanding of pathogenesis of one strain of TBEV in this strain of mice, which should be useful to others working with this pathogen.  I have only two additional comments:

Were the virus inocula used on the mice backtitrated to confirm dose.  I’ve always found it a bit scary to assume that such low doses were actually delivered, and it would not be surprising if what was inoculated was 2 or even 5-fold different that the calculations.  Some assurance by the authors of confidence in dose would be good to include.

Was there a reason that the PCR results presented for viremia (Table 3) were not quantitated but rather presented and positive on negative.  Is quantitative data available – if so, it would be useful to include.  Similarly, for the organ burdens of virus, all of the values are presented as copies per ul when the typical values are per gram; this is not a major criticism.

Author Response

Dear Reviewer,

First, we would like to thank you for your time and your thorough review of the manuscript. Please find our answer to your comments below.

Comment: " Were the virus inocula used on the mice backtitrated to confirm dose.  I’ve always found it a bit scary to assume that such low doses were actually delivered, and it would not be surprising if what was inoculated was 2 or even 5-fold different that the calculations.  Some assurance by the authors of confidence in dose would be good to include."

We appreciate your feedback on our TBEV article and your suggestion regarding back titration of the virus inocula used in the mice to confirm the dose. Unfortunately, we did not back-titrate the virus inocula at the beginning of this experiment, as it is not something people use to do. Indeed, after viral production, the virus was frozen and then one aliquot was defrozen before being titrated as usual. Then for this experiment, one aliquot was defrozen, the different dilutions were performed and used directly. Nevertheless, we agree this is a very good advice and we will do it for further experiment. We have added one line of comment on this aspect in the Discussion section, line 392.

Comment: " Was there a reason that the PCR results presented for viremia (Table 3) were not quantitated but rather presented and positive on negative. Is quantitative data available – if so, it would be useful to include.  Similarly, for the organ burdens of virus, all of the values are presented as copies per ul when the typical values are per gram; this is not a major criticism".

The idea by checking the presence of virus in the blood of the mice was to evaluate the number of positive mice per group at different time points, not to quantify the amount of virus in the blood at different time point. Unfortunately, we didn’t quantify the number of copy/µL of virus in the blood of the mice. .

Regarding the use of "copies per ul" for organ loads instead of the more typical "copies per gram", we acknowledge your observation. Unfortunately, we didn’t weigh the organs, so we were able to estimate the copy/µL according of the number of microliter of RNA analyzed and then to the number of microliter for the entire organ. Moreover, usually with this technic of digital quantification, other authors also express their result in copy/µl.